# Impact of Different Surface Treatments on Shear Bond Strength between Two Zirconia Ceramics and a Composite Material

**DOI:** 10.3390/bioengineering11101003

**Published:** 2024-10-07

**Authors:** Se-Hyoun Kim, Young-Jun Lim, Dae-Joon Kim, Myung-Joo Kim, Ho-Boem Kwon, Yeon-Wha Baek

**Affiliations:** 1Department of Prosthodontics and Dental Research Institute, School of Dentistry, Seoul National University, Seoul 03085, Republic of Korea; speedluss@snu.ac.kr (S.-H.K.); silk1@snu.ac.kr (M.-J.K.);; 2VASIC Research Center, Department of Dentistry, School of Dentistry and Dental Research Institute, Seoul National University, Seoul 03085, Republic of Korea; 3Department of Prosthodontics, Gwanak Center, Seoul National University Dental Hospital, Seoul 08826, Republic of Korea; obero7@snu.ac.kr

**Keywords:** zirconia ceramics, adhesion, plasma treatment, shear bond strength

## Abstract

The purpose of this study was to compare the surface changes and shear bond strength between a resin composite and two zirconia ceramics subjected to sandblasting and forming gas (5% H_2_ in N_2_) plasma surface treatment. Two types of zirconia ceramic specimens (3Y-TZP and (Y,Nb)-TZP) were divided into groups based on the following surface treatment methods: polishing (Control), sandblasting (SB), sandblasting and plasma (SB-P), and plasma treatment (P). Subsequently, chemical surface modification was performed using Clearfil SE Bond (Kuraray, Tokyo, Japan), and the Filtek Z-250 (3M, Maplewood, MN, USA) resin composite was applied. Shear bond strengths (SBS) and surface characteristics were determined. Plasma treatment was effective in increasing the wettability. For SBS, there were significant differences among the groups, and the (Y,Nb)-TZP and SB-P groups showed the highest bond strength. Similarly, for the 3Y-TZP specimens, the shear bond strength increased with both plasma and sandblasting treatments, although no statistically significant change was observed. In the P group, both (Y,Nb)-TZP and 3Y-TZP showed a significant decrease in shear bond strength with the resin composite compared to the control group.

## 1. Introduction

Zirconia ceramics are widely used in dental restorations because of their biocompatibility, high mechanical strength, low thermal conductivity, and excellent esthetics [1]. In particular, zirconia ceramics fabricated using CAD/CAM technology are increasingly used owing to their favorable physical properties and ease of manufacture [2].

For implant superstructures, interproximal contact loss can occur at an average of 1.9–3.6 years following prosthesis placement due to the movement of adjacent teeth. The cumulative interproximal contact loss rate was 80.8% 5 years after prosthesis delivery [3]. Therefore, a post-restoration repair is required to fill the space resulting from interproximal contact loss to prevent food impaction. Dental resin composites can be used for intraoral and extraoral repairs in a single visit. However, the low bond strength between zirconia ceramics and resin composites often leads to frequent debonding [2]. Zirconia ceramics possess low surface energy, which limits their adhesion to resin composites even after physical and chemical surface treatments, such as acid etching, sandblasting, and primer application [4,5,6,7,8]. Therefore, various physical and chemical methods have been studied to improve the adhesive properties of zirconia ceramics [9,10]. Sandblasting has been reported to remove contaminants from the zirconia ceramics surface and increase surface roughness, thereby improving the mechanical bonding strength [11]. Additionally, 10-methacryloyloxydecyl dihydrogen phosphate (MDP), a primer that mediates chemical bonding between zirconia ceramics and resin composites, has been shown to provide superior bond strength compared to conventional phosphate monomers [12] and to maintain long-term adhesion between zirconia ceramics and resin composites [13].

Furthermore, recent studies have focused on using plasma treatment to remove contaminants from zirconia ceramics surfaces and modify their surface properties, thereby enhancing their adhesion to resin composites [14,15,16,17,18]. Plasma treatment has been reported to add hydrophilic base groups to inert materials, such as zirconia ceramics, and organic materials, such as resin composites [19]. Zirconia ceramics surfaces have a low concentration of hydroxyl groups, which results in a hydrophobic surface. Plasma treatment can reduce carbon-based contaminants and modify the surface, thereby enhancing hydrophilicity [20]. Other studies have reported that plasma can add hydroxyl groups to the Y-TZP surface, promoting the formation of intermolecular secondary forces such as van der Waals bonds, which contribute to the bonding strength with resin composites [18,21]. By applying plasma treatment to zirconia ceramics, the surface polarity and surface energy can be increased, thereby enhancing the bonding strength with the resin composite using a primer [18,22]. However, numerous studies have indicated that plasmas, such as oxygen, argon, and nitrogen plasmas, do not significantly impact the clinical bonding strength between zirconia ceramics and resin composites [23,24,25,26,27,28,29].

The surfaces of zirconia ceramics are typically composed of zirconium dioxide, where the oxygen atoms on the surface mediate the chemical bonding between the oxo group of the primer (P = O) and the zirconia ceramics. However, it has been reported that the bonding strength between reduced zirconia ceramics cations (Zr^4+^) and the oxo group (P = O) is stronger [30]. In this context, studies have been conducted to reduce the metal oxide layer on a surface using plasma treatment with a forming gas (a mixture of hydrogen and nitrogen gasses) [31,32]. However, the application of this method to zirconia ceramics has not yet been reported.

Recently, (Y,Nb)-TZP, a type of zirconia ceramic with added yttria and niobium, was introduced. This material reduces the number of oxygen vacancies within the lattice, thereby decreasing the oxygen diffusion rate and achieving electrical neutrality. This resulted in reduced low-temperature degradation. Furthermore, (Y,Nb)-TZP is recognized for its high fracture toughness and excellent plasticity [33], making it an effective material for milling crowns using CAD/CAM technology, even in a fully sintered state.

The purpose of this in vitro study was to compare two zirconia ceramics surface changes and the shear bond strength of the resin composite when these ceramics were subjected to sandblasting and forming gas (5% H_2_ in N_2_) plasma surface treatment. The null hypothesis of this study was as follows: plasma surface treatment would not alter the zirconia ceramic surface properties or the shear bond strength of the resin composite.

## 2. Materials and Methods

### 2.1. Specimen Preparation

Two types of zirconia ceramic specimens (3Y-TZP and (Y,Nb)-TZP, provided by Vatech MCIS, Hwaseong, Republic of Korea) were prepared as cylindrical disks with a diameter of 15 mm and a thickness of 2 mm. The specimens were divided into groups based on the experimental design (Figure 1). The sample size was determined based on data from past publications [9,17,21] and available resources. No power analysis was performed. The specimen surfaces were polished using 800-grit and 1000-grit silicon carbide abrasive papers in distilled water, ultrasonically cleaned in 95% alcohol for 5 min, and subsequently dried.

Sandblasting was then performed using 50 µm aluminum oxide particles at a pressure of 3 bar (0.3 MPa) for 5 s from a distance of 5 mm. After sandblasting, the specimens were ultrasonically cleaned with 95% ethanol for 2 min, dried, and stored.

### 2.2. Plasma Treatment

Plasma surface treatment was conducted using a plasmatreat device (Plasmatreat, Steinhagen, Germany) under the conditions specified in Table 1, with a forming gas (5% H_2_ in N_2_) [31] applied at atmospheric pressure and room temperature (Figure 2). The exposure time for both types of zirconia ceramic specimens was set at 2 s to prevent discoloration (Figure 3).

### 2.3. Priming

Following plasma treatment, chemical surface modification was performed using Clearfil SE Bond (Kuraray, Tokyo, Japan), which has been reported to effectively increase the bond strength between zirconia ceramics and resin composites [34]. According to the manufacturer’s instructions, a single layer of primer was applied and gently air-dried, followed by the application of an adhesive with subsequent gentle air-drying and light curing for 10 s.

### 2.4. Resin Composite Application

Subsequently, the Filtek Z-250 (3M, Maplewood, MN, USA) resin composite was applied using an acrylic straw with an internal diameter of 3 mm and a thickness of 4 mm. Excess material was removed with a sharp knife, and light-curing was performed for 40 s in four directions (1200 mW/cm^2^). The acrylic straw was slowly removed and compressed using a condenser.

### 2.5. Analysis

The specimens were stored in distilled water at room temperature for 24 h (ISO 29022) to allow hydrolytic degradation of the interfacial components via static water storage aging.

The shear bond strength was measured with the force applied by the knife-edge shearing rod aligned with the bonding interface. ISO TR 11405 recommends applying force at a speed of 1 mm/min (Figure 4).

Additional specimens were prepared for surface energy analysis via contact angle analysis and SEM (Scanning Electron Microscope) surface measurements. Surface roughness analysis was performed using confocal laser scanning microscopy (CLSM; LSM 800; Carl Zeiss, Jena, Germany). The ConfoMap Premium 7.4.8341 software (Carl Zeiss, Jena, Germany) was used to calculate the surface roughness parameters. The resulting 3D surface areas were processed using templates in the ConfoMap software.

To analyze the elemental composition of specimens, Energy Dispersive X-ray Spectroscopy (EDS, Quantax 200 on Apreo S, Karlsruhe, Germany) was performed under the following conditions: high vacuum, 20 kV, 0.40 nA energy range, 10.0 mm working distance, and sample coated with platinum. The EDS data were processed using Bruker Esprit software (version 2.1) (Bruker Nano, Berlin, Germany) to obtain quantitative analysis results.

### 2.6. Statistical Analysis

The shear bond strength data were first analyzed using the Kolmogorov–Smirnov test to determine whether the data had a normal distribution (IBM SPSS Statistics v26; IBM Corp., New York, NY, USA). As the shear bond strength results did not show a normal distribution, the Mann–Whitney U test was used to compare the surface treatment methods. The Kruskal–Wallis test was used for multiple comparisons within each group.

## 3. Results

### 3.1. Contact Angle

Following plasma treatment, the contact angles with both water (indicative of polarity) and diiodomethane (indicative of non-polarity) decreased for the 3Y-TZP specimens (Figure 5).

For (Y,Nb)-TZP, the contact angles with both water (indicative of polarity) and diiodomethane (indicative of non-polarity) decreased after the plasma treatment (Figure 6). Additionally, prior to plasma treatment, (Y,Nb)-TZP exhibited lower contact angles with water and diiodomethane than 3Y-TZP. After the plasma treatment, both zirconia ceramic types exhibited similar contact angles.

### 3.2. Surface Roughness

Ra is the two-dimensional (2D) counterpart of the three-dimensional (3D) descriptor Sa. Both Ra and Sa reflect the arithmetic mean of the absolute values of the surface points departing from the mean plane within the sampling area. This 3D representation provides a more comprehensive understanding of the surface texture by considering the variations in height across the surface in three dimensions rather than just the average roughness [35]. Both types of zirconia ceramics exhibited increases in the 2D and 3D surface roughness parameters (Ra and Sa values) following sandblasting. Subsequent plasma treatment resulted in a decrease in the Ra and Sa values for both types of zirconia ceramics (Table 2, Figure 7).

### 3.3. Scanning Electron Microscope (SEM)

Energy Dispersive X-ray Spectroscopy (EDS, Quantax 200 on Apreo S, Karlsruhe, Germany) showed a decrease in the atomic mass percentage of oxygen (wt%, mass norm.) from 22% to 17.8–18.4% for 3Y-TZP and from 28.2% to 16.5–22.4% for (Y,Nb)-TZP after plasma treatment. In the SEM images, (Y,Nb)-TZP exhibited more pronounced surface changes due to sandblasting than 3Y-TZP. Both specimens displayed a characteristic honeycomb pattern after plasma treatment, with (Y,Nb)-TZP exhibiting a larger particle morphology (Figure 8).

### 3.4. Shear Bond Strength (SBS)

In the case of (Y,Nb)-TZP, unlike 3Y-TZP, the sandblasting and plasma treatment group resulted in a significant increase in shear bond strength compared to the control group (α = 0.04). Furthermore, when sandblasting and plasma treatment were combined, both specimens showed an increased shear bond strength compared to when only sandblasting was performed, with (Y,Nb)-TZP demonstrating a more pronounced increase. Upon sandblasting treatment administration, 3Y-TZP exhibited stronger shear bond strength than (Y,Nb)-TZP. However, when sandblasting was combined with plasma treatment, (Y,Nb)-TZP exhibited a stronger shear bond strength than 3Y-TZP.

For 3Y-TZP, the control group treated only with Clearfil SE Bond exhibited a higher shear bond strength than the (Y,Nb)-TZP group treated with the same surface treatment. When only the plasma surface treatment was applied without sandblasting, both types of zirconia ceramics showed a statistically significant decrease in shear bond strength compared to the other groups (Figure 9 and Table 3).

## 4. Discussion

Based on the findings of this study, plasma surface treatment alters the surface properties of zirconia ceramics and the shear bond strength of resin composites. Consequently, the null hypothesis of this study is rejected.

Both types of zirconia ceramics showed no significant changes in Ra and Sa values when treated with plasma alone, which is consistent with previous findings indicating that plasma treatment does not alter zirconia ceramics’ surface roughness [23,25,36].

However, after both types of zirconia ceramics underwent plasma treatment following sandblasting, a decrease in surface roughness was observed compared with specimens treated only with sandblasting. After plasma treatment, the SEM images (Figure 8) revealed the reappearance of zirconia ceramics particles observed before polishing, indicating that the remodeling of the original surface state was altered by polishing and sandblasting.

Generally, sandblasting increases surface roughness, thereby enhancing micromechanical bonding strength [37]. However, despite the reduction in surface roughness with the subsequent plasma treatment, the shear bond strength increased for both types of zirconia ceramics in our study. This effect was particularly pronounced for (Y, Nb)-TZP. Moreover, a statistically significant increase in shear bond strength was observed compared to that in the control group (Figure 9).

In summary, the results of this study suggest that surface roughness and shear bond strength are not always correlated [38], and despite the reduction in mechanical roughening, plasma treatment can improve the chemical interaction between zirconia ceramics and resin composite.

The primer reacts differently on the zirconia ceramics surface, predominantly through hydrogen bonding between the P = O (oxo group) and Zr-OH groups (Figure 10) [13,30,39]. Pre-treatment with the Clearfil SE Bond increases the number of strong bonds, such as P-O-Zr [30]. Moreover, utilizing forming gas plasma treatment can reduce zirconia ceramics to enhance the frequency of strong ion bonds (P-O-Zr) [31]. According to the EDS results from this experiment, the oxygen atom weight ratio decreased in both types of zirconia ceramics after plasma treatment, and the resulting increase in shear bond strength appeared to be more pronounced in (Y,Nb)-TZP than in 3Y-TZP.

In the case of (Y,Nb)-TZP, the shear bond strength increased when sandblasting was combined with plasma treatment, as evidenced by the SEM images showing pronounced surface changes due to plasma. The larger particle size observed in the SEM images of (Y, Nb)-TZP is related to the initial powder particle size and indicates a higher material toughness [34]. Additionally, (Y,Nb)-TZP exhibits a high ductility of niobium and a higher ratio of tetragonal phases, including a high c-axis to a-axis ratio in the t-phase and a higher content of the t-phase itself, suggesting a greater potential for phase transformation upon surface treatment [40]. Therefore, changes in the strength of (Y,Nb)-TZP should be considered in future studies.

The maximum acceptable amount of the monoclinic phase in zirconia ceramics is 25% [41], and previous studies have indicated that plasma treatment does not induce phase transformation in 3Y-TZP [24,28]. In contrast, the phase transformation of zirconia ceramics to the monoclinic form due to sandblasting has been observed, with reports suggesting an increase of approximately 14–15% in the monoclinic phase volume after sandblasting in 3Y-TZP [42]. The presence of niobium is known to increase the susceptibility to phase transformation compared with 3Y-TZP [43], and these tendencies and phase ratios may contribute to the differing effects of plasma treatment on the two zirconia ceramics types.

He et al. [44] highlighted the negative impact of the remaining alumina particles on adhesion after sandblasting, necessitating thorough cleaning. Silane and MDP present in primers can bond with alumina; however, such bonds have been reported to have low stability against hydrolysis [45,46]. Energy Dispersive Spectrometry analysis in this study showed a decrease in the Al percentage from 0.9% to 0.6% for 3Y-TZP and from 5.5% to 1.1% for (Y,Nb)-TZP after plasma treatment, suggesting that this reduction could contribute to increased hydrolysis stability.

Consistent with previous studies [29,47,48], this study also found that plasma and primer treatments without sandblasting resulted in a significant decrease in the shear bond strength for both types of zirconia ceramics compared to the control group. This suggests that plasma treatment without sandblasting is insufficient to effectively reduce zirconia ceramics, leading to reactions between various surface substances and plasma that produce byproducts that interfere with primer bonding. Additionally, superoxide anion radicals generated by plasma irradiation may penetrate zirconia ceramics micro defects and be converted into gas during the resin composite polymerization process, potentially hindering bonding [24]. Furthermore, some studies have indicated that the increased hydrophilicity from plasma treatment may contribute to degradation at the zirconia ceramics–resin composite adhesive interface [20].

Additionally, an analysis of the contact angles after plasma treatment showed increased surface energy, consistent with previous research, indicating a decrease in water contact angles of approximately 12° [18,26]. However, despite the increase in surface energy, a proportional enhancement of the shear bond strength with the resin composite was not observed. These results were consistent with previous findings [24].

In the case of 3Y-TZP, the control group coated with Clearfil SE Bond without surface treatment exhibited a higher shear bond strength than (Y,Nb)-TZP, suggesting that the surface energy of the primer is more tailored to 3Y-TZP. This correlates with the findings shown in Figure 5, in which the two zirconia ceramics control groups exhibit different surface energies. However, after plasma treatment, both materials showed similar trends in the contact angle and the shear bond strength, indicating that the chemical affinity for the Clearfil SE Bond was homogenized between the two materials due to plasma treatment.

Without sandblasting, universal primers enhance the bond strength between zirconia ceramics and the resin composite more effectively than phosphate-based primers [49]. Given that each primer reacts differently with the zirconia ceramics surface [13], further research is required to select the optimal primer for (Y,Nb)-TZP.

Except for the group treated with plasma alone, the shear bond strength between zirconia ceramics and resin composite exceeded 15 MPa, demonstrating a bond strength that could be clinically viable. Thus, in limited situations, it may be feasible to remove the zirconia ceramic superstructure from the implant and attempt resin repair extra orally. However, as this is an in vitro study, further research is needed on the strength of the resin composite bond and zirconia ceramic restorations that have aged in the mouth. Additionally, because the intraoral conditions, such as moisture, were not replicated in this experiment, further studies considering these factors will be necessary.

To evaluate the extent of zirconia ceramics reduction through plasma treatment, the specimens were vacuum-sealed after plasma treatment and subjected to elemental analyses. Owing to geographical constraints, the process from plasma treatment to elemental analysis required approximately 10 h to complete. To observe the changes over time after plasma treatment, additional elemental analysis conducted two weeks later revealed a slight increase of approximately 1.46 wt% in the mass percentage of oxygen atoms. This is consistent with previous findings indicating time-dependent changes following plasma treatment [50], suggesting that further research is needed to evaluate the effects of storage methods and duration following plasma treatment.

Based on research indicating the potential long-term degradation of bond strength owing to a lack of mechanical bonding [13], additional studies are required to measure the shear bond strength after multiple thermal cycles.

## 5. Conclusions

Within the limitation of this in vitro study.

When sandblasting and plasma treatments were combined in the (Y,Nb)-TZP specimens, the shear bond strength with the resin composite significantly increased compared to the control group.Similarly, for the 3Y-TZP specimens, the shear bond strength increased with both plasma and sandblasting treatments, although no statistically significant change was observed.Without sandblasting, when zirconia ceramic specimens were treated with plasma and primers, both (Y,Nb)-TZP and 3Y-TZP showed a significant decrease in the shear bond strength with the resin composite compared to the control group.Without mechanical pre-treatment, adhesive bonding with the resin composite using the Clearfil SE Bond is more advantageous for 3Y-TZP than for the (Y,Nb)-TZP.

## Figures and Tables

**Figure 1 bioengineering-11-01003-f001:**
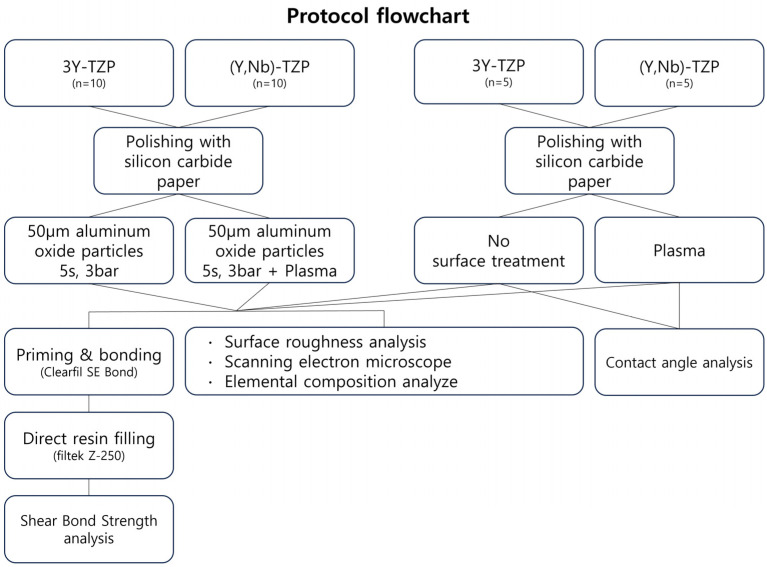
Protocol flowchart.

**Figure 2 bioengineering-11-01003-f002:**
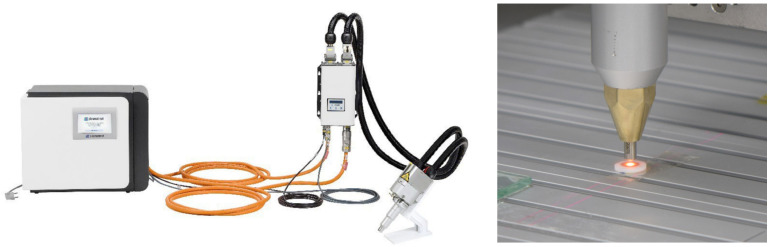
Plasma treatment equipment (Plasmatreat, Steinhagen, Germany).

**Figure 3 bioengineering-11-01003-f003:**
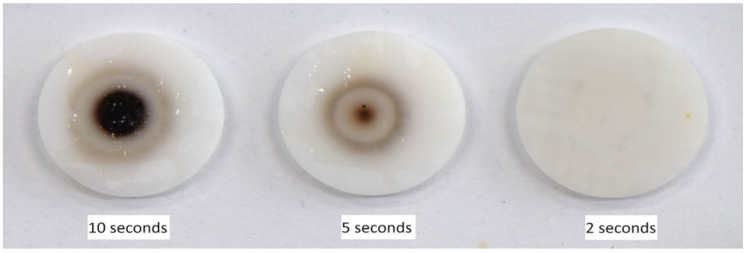
Discoloration patterns according to plasma exposure time (plasma exposures of 10 s, 5 s, and 2 s). Since no discoloration was observed in the 3Y-TZP and (Y,Nb)-TZP specimens after two 2 s of exposure, the experiments were conducted under this condition.

**Figure 4 bioengineering-11-01003-f004:**
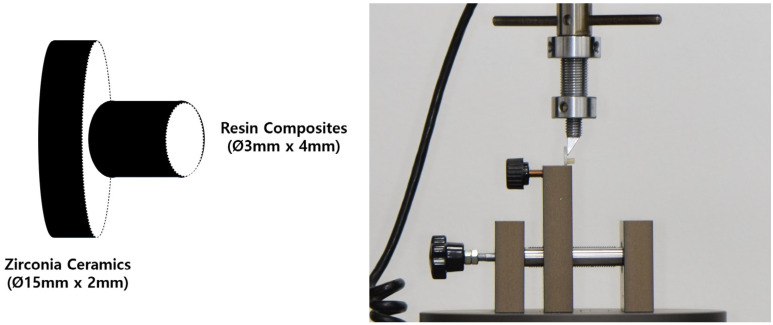
Universal test machine for measuring shear bond strength.

**Figure 5 bioengineering-11-01003-f005:**
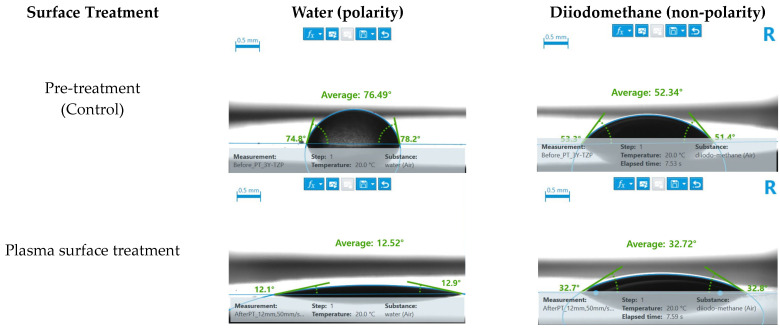
3Y-TZP surface contact angle analysis.

**Figure 6 bioengineering-11-01003-f006:**
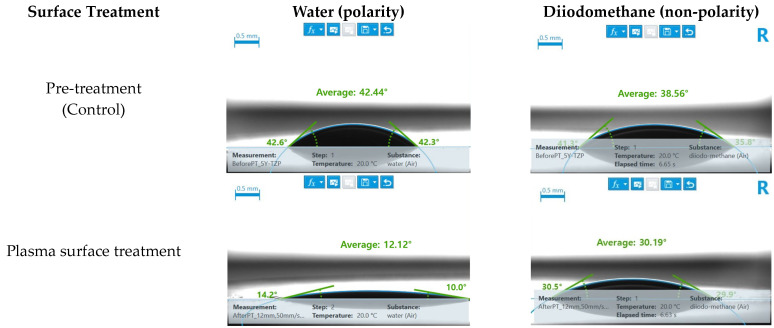
(Y,Nb)-TZP surface contact angle analysis.

**Figure 7 bioengineering-11-01003-f007:**
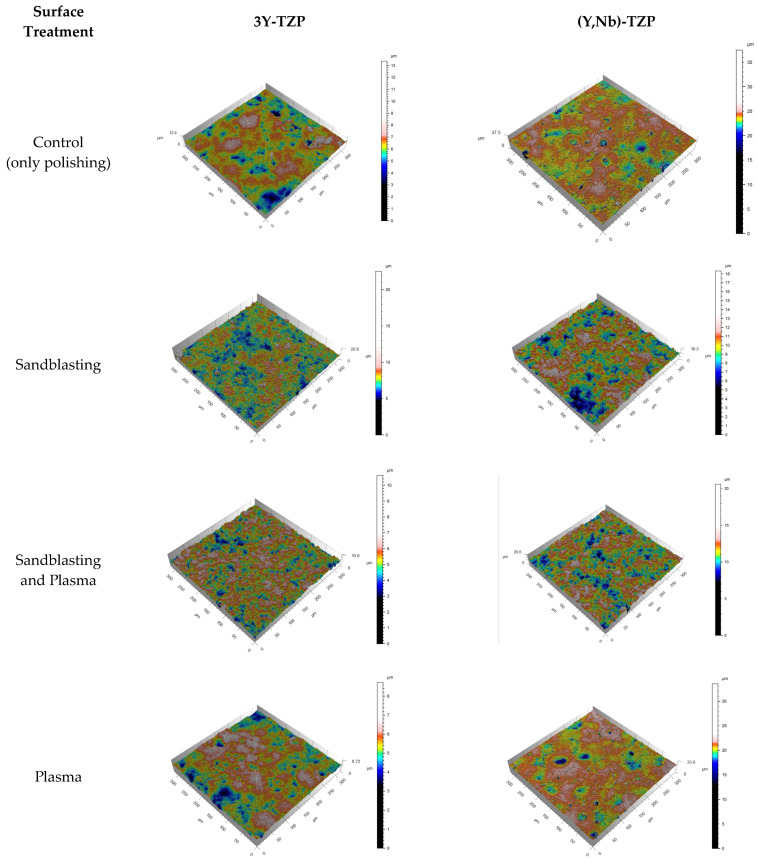
Surface 3D image.

**Figure 8 bioengineering-11-01003-f008:**
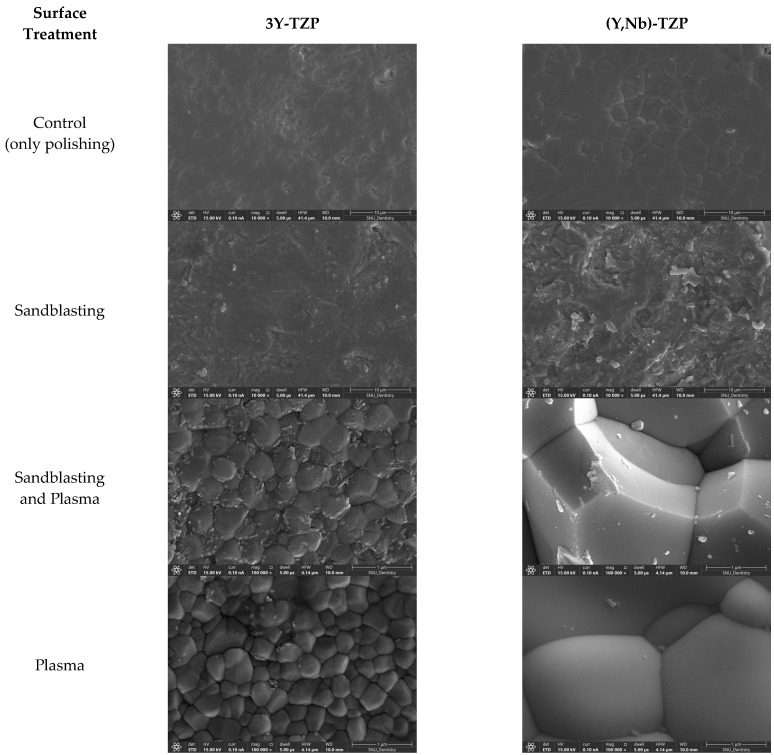
Scanning Electron Microscope image (×10,000).

**Figure 9 bioengineering-11-01003-f009:**
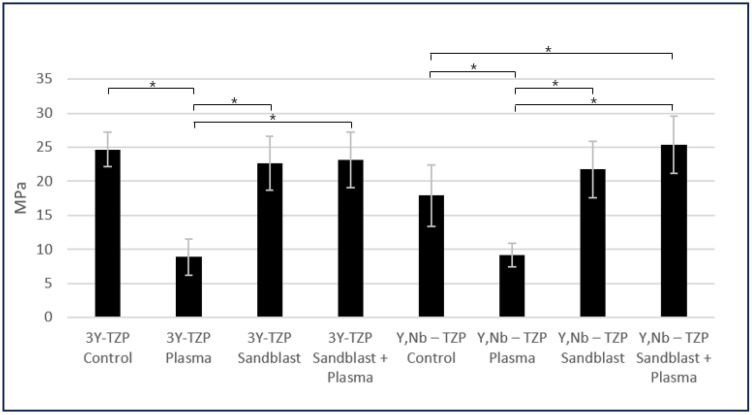
Comparison of shear bond strengths (95% Confidence interval). * Statistical significance (*p* < 0.05).

**Figure 10 bioengineering-11-01003-f010:**
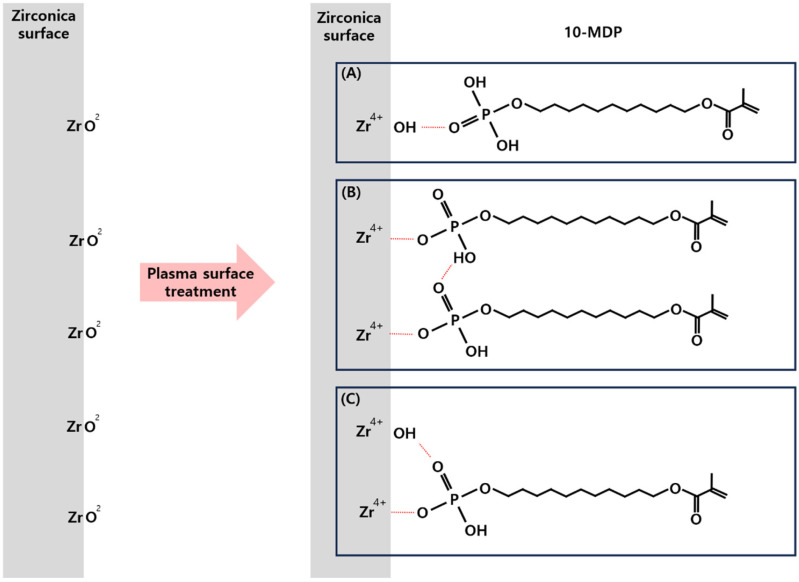
Chemical interaction mechanism of 10-MDP with zirconia ceramics [39] (**A**): Hydrogen bonding between zirconia ceramics (Zr-OH group) and 10-MDP primer (P = O oxo group). (**B**): Ionic bonding between zirconia ceramics (Zr) and 10-MDP primer (P = O oxo group) and hydrogen bonding between zirconia ceramics and 10-MDP. (**C**): Combination of (**A**,**B**) forms.

**Table 1 bioengineering-11-01003-t001:** Plasma treatment conditions.

Description	Parameter Unit
P (Power)	782 W
U (Voltage)	300 V
I (Ampere)	13.8 A
Frequency	23 kHz
Q (Air Quantity)	40 L/m
P (Pressure)	37 mBar

**Table 2 bioengineering-11-01003-t002:** Surface roughness analysis (95% Confidence interval).

	Ra	Rq	Rv	Rz	Sa
3Y-TZP–C	0.35 (±0.04)	1.21 (±0.40)	1.61 (±0.46)	2.82 (±0.86)	0.56 (±0.02)
3Y-TZP–SB	0.64 (±0.04)	2.83 (±0.29)	2.13 (±0.13)	4.96 (±0.41)	0.76 (±0.02)
3Y-TZP–SB + P	0.46 (±0.02)	1.56 (±0.13)	1.70 (±0.17)	3.26 (±0.14)	0.59 (±0.00)
3Y-TZP–P	0.30 (±0.02)	1.03 (±0.07)	1.08 (±0.09)	2.11 (±0.10)	0.52 (±0.03)
(Y,Nb)-TZP–C	0.54 (±0.05)	2.02 (±0.33)	3.41 (±0.82)	5.43 (±0.90)	0.82 (±0.02)
(Y,Nb)-TZP–SB	0.88 (±0.12)	3.20 (±0.59)	2.83 (±0.29)	6.03 (±0.88)	1.32 (±0.12)
(Y,Nb)-TZP–SB + P	0.72 (±0.01)	2.56 (±0.16)	2.84 (±0.08)	5.40 (±0.08)	1.17 (±0.05)
(Y,Nb)-TZP–P	0.62 (±0.11)	3.56 (±0.59)	5.29 (±0.90)	7.85 (±1.20)	0.89 (±0.04)

Abbreviations: Ra: arithmetic average of surface profile; Rq: root mean square roughness; Rv: the average of deepest valleys; Rz: the maximum average of the highest peak and lowest valley; Sa: three-dimensional (3D) descriptor. C: Control (only polishing); SB: Sandblasting; P: Plasma.

**Table 3 bioengineering-11-01003-t003:** Comparison of shear bond strengths (95% Confidence interval).

(Unit: MPa)	Control	Plasma	Sandblast	Sandblast + Plasma
3Y-TZP	24.66 (±2.53) ^A^	8.87 (±2.67) ^ABC^	22.68 (±3.96) ^B^	23.14 (±4.09) ^C^
(Y,Nb)-TZP	17.89 (±4.56) ^ad^	9.17 (±1.79) ^abc^	21.75 (±4.13) ^b^	25.34 (±4.23) ^cd^
*p*-value	0.095	0.529	0.631	0.470

Statistical significance (*p* < 0.05) is indicated by the same superscript letter in the columns.

## Data Availability

Data are contained within the article.

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
