# Peer review of "Impact of Different Surface Treatments on Shear Bond Strength between Two Zirconia Ceramics and a Composite Material"

_bioengineering, 2024, doi:10.3390/bioengineering11101003_

Round 1
Reviewer 1 Report
Comments and Suggestions for Authors
Dear authors,
thank you very much for your manuscript focusing on an important topic in adhesive dentistry. Some improvements should be done prior publication. The style and English language could be improved,
1. Title: Please clarify the title. Suggestion: Impact of Different Surface Treatments on Shear Bond Strength between Two Zirconia Oxide Ceramics and a Composite Material
2. Abstract: Please give more information regarding material and methods, results and conclusions within the abstract to increase reader´s interest. The numbers should be removed. They are not necessary following the instructions for authors.
3. Please improve the keywords. I would recommend not using abbreviations.
4. Introduction: Please use the correct term “zirconia oxide ceramics” or zirconia ceramics” throughout the entire manuscript. Please adjust.
Zirconia ceramics are also used for restoring natural teeth not only implants. Pleas adjust the introduction.
Your study is focusing on the repair of defect zirconia ceramic restauration with composite materials. The prevalence of fractures and the clinical challenges should be addressed in the introduction. Please adjust the introduction to this point. This should also be included in the aim of your study. The difficulties and challenges repairing ceramic restaurations should also be mentioned. Former results should be included.
Please give a clear aim of your study. Please give also a hypothesis of your intensive investigation.
5. Material and Methods:
Please give exact information about the used materials and devices (including names, cities and countries). Sometimes this information is available, sometimes not or incomplete.
Please check the spelling throughout the entire manuscript. Mistakes are present.
Figure 4 is difficult to understand. Please use a higher magnification or a graphical expression. This might help readers. The legend should be adjusted. Thank you.
Did you perform any sample size calculation in advance? Please add this information.
6. Discussion:
Please discuss your hypothesis and null-hypothesis stated at the beginning.
Please discuss the limitations, the advantages and disadvantages of your study in detail. Are there any clinical consequences for reader´s repairing zirconia ceramics? Please discuss the intra- and extraoral possibilities to perform repairs of zirconia ceramics. Please discuss the possible impact of your results on clinical practice. This might increase reader´s interest.
7. Conclusion:
Please weaken the conclusion due to the in vitro design of the investigation. Please
8. Final statements
Some final statements are missing: Please include the section "Author Contribution", give more information about the company “Plasmatreat (Location, Country).
9. References
Please adjust the references to the journal requirements and use the recommended style for all references following the author instructions.
Comments on the Quality of English LanguagePlease check the English language (style and spelling).
Author Response
Comments and suggestions
thank you very much for your manuscript focusing on an important topic in adhesive dentistry. Some improvements should be done prior publication. The style and English language could be improved,
1) Title: Please clarify the title. Suggestion: Impact of Different Surface Treatments on Shear Bond Strength between Two Zirconia Oxide Ceramics and a Composite Material
Response: We appreciate the suggestion from the reviewer. It seemed that the title had an excessively long enumeration regarding surface treatment, so we appreciate and accept the suggestion from the reviewer.
- Title: Impact of Different Surface Treatments on Shear Bond Strength between Two Zirconia Ceramics and a Composite Material
2) Abstract: Please give more information regarding material and methods, results and conclusions within the abstract to increase reader´s interest. The numbers should be removed. They are not necessary following the instructions for authors.
Response: Thank you for your suggestion. In accordance with the author guidelines, we have removed the numbering and added the following information regarding materials and methods, results, and conclusions.
- Line 12-24: Abstract: The purpose of this study was to compare the surface changes and shear bond strength between two Zirconia ceramics and resin composites subjected to sandblasting and forming gas (5% H2 in N2) plasma surface treatment. Two types of Zirconia ceramic specimens (3Y-TZP and (Y,Nb)-TZP) were divided into groups based on the following surface treatment methods: polishing (Control), sandblasting (SB), sandblasting and plasma (SB-P), and plasma treatment (P). Subsequently, chemical surface modification was performed using Clearfil SE Bond (Kuraray, Tokyo, Japan) and the Filtek Z-250 (3M, MN, US) resin composites was applied. Shear bond strengths (SBS) and surface characteristics were determined. Plasma treatment was effective in increasing the wettability. For SBS, there were significant differences among the groups and the (Y,Nb)-TZP & SB-P group showed the highest bond strength. Similarly, for the 3Y-TZP specimens, the shear bond strength increased with both plasma and sandblasting treatments, although no statistically significant change was observed. In the P group, both (Y,Nb)-TZP and 3Y-TZP showed a significant decrease in shear bond strength with the resin composites compared to the control group.
3) Please improve the keywords. I would recommend not using abbreviations.
Response: Thank you. We have deleted the abbreviation for zirconia and added "Zirconia Ceramics" as a new keyword
- Line 6: Keywords: Zirconia ceramics; Adhesion; Plasma treatment; Shear bond strength
4) Introduction: Please use the correct term “zirconia oxide ceramics” or zirconia ceramics” throughout the entire manuscript. Please adjust.
4-1) Zirconia ceramics are also used for restoring natural teeth not only implants. Pleas adjust the introduction
4-2) Your study is focusing on the repair of defect zirconia ceramic restauration with composite materials. The prevalence of fractures and the clinical challenges should be addressed in the introduction. Please adjust the introduction to this point. This should also be included in the aim of your study. The difficulties and challenges repairing ceramic restaurations should also be mentioned. Former results should be included.
4-3) Please give a clear aim of your study. Please give also a hypothesis of your intensive investigation.
Response: We thanks for the reviewer for pointing out this issue. We’ve checked the subscript throughout the paper and revised "zirconia" to "zirconia ceramics."
4-1) As you mentioned, We have modified the text to specify that zirconia ceramics is used not only for implants but also as a dental restorative material that includes natural tooth restoration.
- Line 29-32: Zirconia ceramics is widely used in dental restorations because of its biocompatibility, high mechanical strength, low thermal conductivity, and excellent esthetics. In particular, Zirconia ceramics fabricated using CAD/CAM technology are increasingly used owing to their favorable physical properties and ease of manufacture.
4-2) Fracture of zirconia ceramics is rarely reported in most studies, so we have removed that section. However, for implants, interproximal contact loss occurs at a significantly high rate of 80.8% within 5 years post-restoration, leading to discomfort from food impaction. As a result, repairs to the zirconia ceramics are often necessary. To emphasize the clinical relevance of this study, as per your suggestion, We have added data on the frequency of such occurrences as follows.
- Line 33-37: For implant superstructures, interproximal contact loss can occur at an average of 1.9–3.6 years following prosthesis placement due to the movement of adjacent teeth. The cumulative interproximal contact loss rate was 80.8% at 5 years after prosthesis delivery. Therefore, a post-restoration repair is required to fill the space resulting from interproximal contact loss to prevent food impaction.
4-3) We have revised the objectives and hypothesis of the study to make them clearer, as follows.
- Line 79-83: The purpose of this in vitro study was to compare two zirconia ceramics surface changes and shear bond strength to the resin composites when zirconia ceramics was subjected to sandblasting and forming gas (5% H2 in N2) plasma surface treatment. The null hypothesis of this study was as follows: Plasma surface treatment would not alter the zirconia ceramic surface properties and shear bond strength to the resin composites.
5) Material and Methods: Please give exact information about the used materials and devices (including names, cities and countries). Sometimes this information is available, sometimes not or incomplete Please check the spelling throughout the entire manuscript. Mistakes are present. Figure 4 is difficult to understand. Please use a higher magnification or a graphical expression. This might help readers. The legend should be adjusted. Thank you. Did you perform any sample size calculation in advance? Please add this information.
Response: We appreciate and accept the suggestion from the reviewer. we have revised the materials and equipment used in this paper to include the names, cities, and countries, as requested. Additionally, We conducted a thorough spelling check throughout the document. Figure 4 illustrates zirconia ceramics was fixed at the bottom with screws, and shear bond strength was measured by applying force with a knife-edge at the resin-zirconia adhesive interface. As per your suggestion, we have created an image of the sample specifications and a schematic diagram and added them. The sample size was determined based on previous studies, and no additional power test was conducted. we have incorporated the related details as follows.
- figure 4:
- Line 94-95: The sample size was determined based on data from past publications and available resources. No power analysis was performed.
6) Discussion:
Please discuss your hypothesis and null-hypothesis stated at the beginning.
Please discuss the limitations, the advantages and disadvantages of your study in detail. Are there any clinical consequences for reader´s repairing zirconia ceramics? Please discuss the intra- and extraoral possibilities to perform repairs of zirconia ceramics. Please discuss the possible impact of your results on clinical practice. This might increase reader´s interest.
Conclusion: Please weaken the conclusion due to the in vitro design of the investigation. Please
Response: We appreciate the suggestion and question from the reviewer. We have stated the null hypothesis for the two zirconia materials at the end of the introduction and described the corresponding results at the beginning of the discussion, as follows.
- Line 79-83: The purpose of this in vitro study was to compare two zirconia ceramics surface changes and shear bond strength to the resin composites when zirconia ceramics was subjected to sandblasting and forming gas (5% H2 in N2) plasma surface treatment. The null hypothesis of this study was as follows: Plasma surface treatment would not alter the zirconia ceramic surface properties and shear bond strength to the resin composites.
- Line 234-236: Based on the findings of this study, plasma surface treatment alters the surface properties of zirconia ceramics and shear bond strength to resin composites. Consequently, the null hypothesis of this study is rejected.
Through this study, it was confirmed that when zirconia ceramic surfaces are treated with sandblasting, plasma, and primer, a shear bond strength with resin that can be clinically attempted is achieved. Additionally, in the case of 3Y-TZP, even without sandblasting, clinically viable results were obtained through priming alone. However, this experiment did not test zirconia that had aged in the mouth, and it has limitations in fully replicating clinical conditions such as intraoral moisture. As a result, this method may not yet be reliably used in the oral cavity and could be attempted for extraoral repair in urgent situations requiring fast repairs. We have also included this point as outlined below, and, as you suggested, We emphasized that this study is an in vitro experiment, requiring cautious interpretation, which has led to a weakening of the conclusions.
- Line 323-330: Except for the group treated with plasma alone, the shear bond strength between zirconia ceramics and resin composites exceeded 15 MPa, demonstrating a bond strength that could be clinically viable. Thus, in limited situations, it may be feasible to remove the zirconia ceramic superstructure from the implant and attempt resin repair extraorally. However, as this is an in-vitro study, further research is needed on the resin composites bond strength to zirconia ceramic restorations that have aged in the mouth. Additionally, because the intraoral conditions, such as moisture, were not replicated in this experiment, further studies considering these factors will be necessary.
8) Final statements Some final statements are missing: Please include the section "Author Contribution", give more information about the company “Plasmatreat (Location, Country).
Response: We thanks for the reviewer for pointing out this issue. we have added the "Author Contribution" section and included the location, city, and country for Plasmatreat.
9) References: Please adjust the references to the journal requirements and use the recommended style for all references following the author instructions.
Response: We thanks for the reviewer for pointing out this issue. We’ve revised all the references according to the author instructions.

Reviewer 2 Report
Comments and Suggestions for Authors
Please indicate that the chemical symbols are for the zirconia specimen.
Fig 2 should plasmatreat be capitalized?
Line 27 please state that these are for dental restorations.
Author Response
Comments and suggestions
1) Please indicate that the chemical symbols are for the zirconia specimen.
Response: We appreciate the suggestion from the reviewer. We utilized two types of zirconia ceramic specimens: (Y,Nb)-TZP and 3Y-TZP. Since the composition of each element varies by manufacturer, the company names have been specified. Additionally, the chemical symbols have been expanded and included as follows, along with an explanation of the characteristics of the specimens.
- Line 73-78: Recently, (Y,Nb)-TZP, a type of Zirconia ceramics with added yttria and niobium was introduced. This material reduces the number of oxygen vacancies within the lattice, thereby decreasing the oxygen diffusion rate and achieving electrical neutrality. This resulted in reduced low-temperature degradation. Furthermore, (Y,Nb)-TZP is recognized for its high fracture toughness and excellent plasticity, making it an effective material for milling crowns using CAD/CAM technology, even in a fully sintered state.
2) Fig 2 should plasmatreat be capitalized?.
Response: Thank you. As you pointed out, Figure 2 needed to start with a capital letter, and that section has now been revised accordingly.
3) Line 27 please state that these are for dental restorations.
Response: As you mentioned, We have modified the text to specify that zirconia ceramics is used not only for implants but also as a dental restorative material that includes natural tooth restoration.
- Line 29-32: Zirconia ceramics is widely used in dental restorations because of its biocompatibility, high mechanical strength, low thermal conductivity, and excellent esthetics. In particular, Zirconia ceramics fabricated using CAD/CAM technology are increasingly used owing to their favorable physical properties and ease of manufacture.

Reviewer 3 Report
Comments and Suggestions for Authors
The purpose of this study was to compare the surface changes and shear bond strength (Y, Nb)-TZP to a composite resin subjected plasma surface treatment. The title of the manuscript is: Impact of Surface Treatments including Sandblasting, Plasma, 2 and Primer on Shear Bond Strength between Zirconia (3Y-TZP 3 & (Y,Nb)-TZP) and Resin Composite, so I would suggest to put in the aim of study both zirconia types and also include the description of materials in the Introduction. The sample size analysis is missing. I would suggest adding Statistical analysis section at the and of Material and methods and to add the limitations of the study.
Author Response
Comments and suggestions
1) The purpose of this study was to compare the surface changes and shear bond strength (Y, Nb)-TZP to a composite resin subjected plasma surface treatment. The title of the manuscript is: Impact of Surface Treatments including Sandblasting, Plasma, 2 and Primer on Shear Bond Strength between Zirconia (3Y-TZP 3 & (Y,Nb)-TZP) and Resin Composite, so I would suggest to put in the aim of study both zirconia types and also include the description of materials in the Introduction.
Response: We appreciate and accept the suggestion from the reviewer. We have revised the objectives and hypothesis of the study to make them clearer, as follows.
- Line 79-83: The purpose of this in vitro study was to compare two zirconia ceramics surface changes and shear bond strength to the resin composites when zirconia ceramics was subjected to sandblasting and forming gas (5% H2 in N2) plasma surface treatment. The null hypothesis of this study was as follows: Plasma surface treatment would not alter the zirconia ceramic surface properties and shear bond strength to the resin composites.
- Line 234-236: Based on the findings of this study, plasma surface treatment alters the surface properties of zirconia ceramics and shear bond strength to resin composites. Consequently, the null hypothesis of this study is rejected.
2) The sample size analysis is missing.
Response: We thanks for the reviewer for pointing out this issue. The sample size was determined based on previous studies, and no additional power test was conducted. we have incorporated the related details as follows.
- Line 94-95: The sample size was determined based on data from past publications and available resources. No power analysis was performed.
3) I would suggest adding Statistical analysis section at the and of Material and methods
Response: We appreciate the suggestion. We have added the statistical analysis section, dividing it as follows.
- Line 155-160:
2.6. Statistical analysis
The shear bond strength data were first analyzed using the Kolmogorov–Smirnov test to determine whether the data had a normal distribution (IBM SPSS Statistics v26; IBM Corp., NY, USA). As the shear bond strength results did not show a normal distribution, the Mann–Whitney U test was used to compare the surface treatment methods. The Kruskal–Wallis test was used for multiple comparisons within each group.
4) and to add the limitations of the study.
Response: We appreciate the suggestion and question from the reviewer. Through this study, it was confirmed that when zirconia ceramic surfaces are treated with sandblasting, plasma, and primer, a shear bond strength with resin that can be clinically attempted is achieved. Additionally, in the case of 3Y-TZP, even without sandblasting, clinically viable results were obtained through priming alone. However, this experiment did not test zirconia that had aged in the mouth, and it has limitations in fully replicating clinical conditions such as intraoral moisture. As a result, this method may not yet be reliably used in the oral cavity and could be attempted for extraoral repair in urgent situations requiring fast repairs. We have also included this point as outlined below, and, as you suggested, we emphasized that this study is an in vitro experiment, requiring cautious interpretation, which has led to a weakening of the conclusions.
- Line 323-330: Except for the group treated with plasma alone, the shear bond strength between zirconia ceramics and resin composites exceeded 15 MPa, demonstrating a bond strength that could be clinically viable. Thus, in limited situations, it may be feasible to remove the zirconia ceramic superstructure from the implant and attempt resin repair extraorally. However, as this is an in-vitro study, further research is needed on the resin composites bond strength to zirconia ceramic restorations that have aged in the mouth. Additionally, because the intraoral conditions, such as moisture, were not replicated in this experiment, further studies considering these factors will be necessary.

Round 2
Reviewer 1 Report
Comments and Suggestions for Authors
Thank you for adressing all points of the first review round.The manscript is improved.
The Author Contribution is added Please use the recommended journal style just using capital letters instead of full names. Please check.
Please check the references. I think now the are all numbered as 1. Please check and correct.
Comments on the Quality of English Language
Minor editing is still necessary.